# Estrogen-Related Receptor α: A Key Transcription Factor in the Regulation of Energy Metabolism at an Organismic Level and a Target of the ABA/LANCL Hormone Receptor System

**DOI:** 10.3390/ijms25094796

**Published:** 2024-04-27

**Authors:** Sonia Spinelli, Maurizio Bruschi, Mario Passalacqua, Lucrezia Guida, Mirko Magnone, Laura Sturla, Elena Zocchi

**Affiliations:** 1Laboratory of Molecular Nephrology, IRCCS Istituto Giannina Gaslini, Via Gerolamo Gaslini 5, 16147 Genova, Italy; 2Section Biochemistry, Department of Experimental Medicine (DIMES), University of Genova, Viale Benedetto XV, 1, 16132 Genova, Italy; mario.passalacqua@unige.it (M.P.); l.guida@unige.it (L.G.); mirko.magnone@unige.it (M.M.); laurasturla@unige.it (L.S.)

**Keywords:** mitochondrial function, biogenesis and proton gradient, energy production, oxphos uncoupling, oxidative stress, AMPK/PGC-1α/SIRT1, neurodegeneration, cardioprotection, skeletal muscle, kidney, BAT

## Abstract

The orphan nuclear receptor ERRα is the most extensively researched member of the estrogen-related receptor family and holds a pivotal role in various functions associated with energy metabolism, especially in tissues characterized by high energy requirements, such as the heart, skeletal muscle, adipose tissue, kidney, and brain. Abscisic acid (ABA), traditionally acknowledged as a plant stress hormone, is detected and actively functions in organisms beyond the land plant kingdom, encompassing cyanobacteria, fungi, algae, protozoan parasites, lower Metazoa, and mammals. Its ancient, cross-kingdom role enables ABA and its signaling pathway to regulate cell responses to environmental stimuli in various organisms, such as marine sponges, higher plants, and humans. Recent advancements in understanding the physiological function of ABA and its mammalian receptors in governing energy metabolism and mitochondrial function in myocytes, adipocytes, and neuronal cells suggest potential therapeutic applications for ABA in pre-diabetes, diabetes, and cardio-/neuroprotection. The ABA/LANCL1-2 hormone/receptor system emerges as a novel regulator of ERRα expression levels and transcriptional activity, mediated through the AMPK/SIRT1/PGC-1α axis. There exists a reciprocal feed-forward transcriptional relationship between the LANCL proteins and transcriptional coactivators ERRα/PGC-1α, which may be leveraged using natural or synthetic LANCL agonists to enhance mitochondrial function across various clinical contexts.

## 1. Introduction

### 1.1. ERRs: Protein Family and Tissue Expression

In 1988, two distinctive nuclear receptors exhibiting conserved features of steroid hormone receptors were identified: estrogen-related receptors α (ERRα) and β (ERRβ) [1]. Ten years later, researchers discovered the third receptor isoform, estrogen-related receptor γ (ERRγ) [2]. The ERR family displays significant sequence similarity with the DNA-binding domain (DBD) and ligand-binding domain (LBD) of estrogen receptor ERα [3]. Despite this resemblance, ERRs cannot bind estrogen or its derivatives, leading to their classification as orphan nuclear receptors [4,5]. In human tissues, ERRα lacks known splice variants, ERRβ has three splice variants, and ERRγ has two splice variants [6]. These splice variants contribute significantly to functional diversity in the proteome [7]. However, a comprehensive understanding of these splice variants suffers from the limited research available [8,9]. The molecular architecture of ERRs closely resembles that of other steroid nuclear receptors, consisting of six conserved regions (A/B, C, D, E/F domains) [10] (Figure 1).

The N-terminal region, known as the A/B domain or activation function-1, is characterized by ligand-independent transcriptional activation. The A/B domains of ERRs contain conserved motifs susceptible to regulation through post-translational modifications, such as SUMOylation and phosphorylation [11,12]. The central C domain, referred to as the DNA-binding domain, houses two highly conserved zinc finger motifs capable of binding to a specific DNA sequence (TCAAGGTCA) known as the ERR response element [13]. ERRs can form monomers, homodimers, or heterodimers with different isoforms [14,15], allowing distinct isoforms to target the same gene due to virtually identical C domains [16]. The D domain acts as a flexible hinge region, providing protein flexibility during DNA binding, and links the C and E regions [17,18]. The E/F domain serves as the ligand-binding domain (LBD), sharing 30-40% homology with the LBD of ERα. However, ERRs cannot bind estrogen or its derivatives due to the absence of critical Cys residues [19]. The LBD contains a conserved helix motif called AF-2 exposed in ERRs. The three ERR isoforms are constitutively active because their E/F domains can bind co-regulators without ligand binding [20,21,22]. Nevertheless, the quest for the natural ligand(s) of ERRs is open, and recent reports indicate that cholesterol may be an important endogenous agonist of ERRα [23,24].

ERRs are essentially regulators of mitochondrial energy metabolism and, as such, they play key roles in various physiological functions that directly or indirectly provide or depend on energy metabolism, mitochondrial biogenesis, oxidative phosphorylation (oxphos), glucose and lipid oxidation, cell proliferation and differentiation, and tumor growth and dissemination [23,25,26,27,28]. Their expression is ubiquitous in the human body (https://www.proteinatlas.org/ENSG00000173153-ESRRA/tissue accessed on 23 April 2024), and they are particularly high in tissues with high energy expenditure or intense metabolic demands [29]. In the same tissues, the expression of peroxisome proliferator-activated receptor Gamma Coactivator 1-alpha (PGC-1α), another transcriptional coactivator, is also particularly high, suggesting a functional link between ERRα and PGC-1α. In adults, ERRα exhibits the highest expression level, ERRγ shows an intermediate level of expression, and ERRβ displays the lowest expression [30]. 

### 1.2. ERRα: Master Regulator of Mitochondrial Biogenesis, Oxidative Phosphorylation, and Energy Production

ERRα, the most extensively studied member of the ERR family, plays a crucial role in various functions related to energy metabolism, particularly in tissues with high energy demands, such as the heart, skeletal muscle, adipose tissue, kidney, and brain. A synthetic summary of its functions is presented in Section 2 of this review. In addition, ERRα also plays important roles in cancer cell metabolism, antioxidant defense (also briefly summarized below), and bone and immune cell metabolism, fields of research that are covered by other reviews [31,32]. In mitochondria, the “power centrals” of cells, metabolic activity is strictly controlled to meet the varying energy demands of cells under different physiological conditions. PGC-1α and nuclear receptor corepressor 1 (NCOR1) are well-known inducers of mitochondrial oxidative metabolism, abundantly expressed in high-energy demand tissues like the heart, skeletal muscle, and brown adipose tissue (BAT). As both lack DNA binding activity, they rely on interactions with other transcription factors to directly bind and regulate downstream target genes. A demonstration of the functional collaboration between ERRα and PGC-1α in regulating energy metabolism in cardiomyocytes and in skeletal myotubes came from a study by Huss et al. [33], who used ERRα overexpressing rat neonatal cardiomyocytes and skeletal muscle cells from ERRα/mice and showed that ERRα is responsible for the mitochondrial metabolic and respiratory effects downstream of PGC-1α activation [33]. ERRs have been subsequently confirmed as key transcription factors that regulate mitochondrial oxidative metabolism and induce PGC-1α and NCOR1 expression [34]. Studies demonstrate that ERRs also play a crucial role in regulating the expression of glycolytic genes, including phosphofructokinase, hexokinase 2, glyceraldehyde phosphate dehydrogenase, and enolase 1, essential components of cytoplasmic glucose metabolism [35]. 

### 1.3. Functional Collaboration between PGC-1α and ERRα: The PGC-1α/ERRα Axis 

According to the BioGRID database (https://thebiogrid.org accessed on 23 April 2024), which compiles protein-protein interactions from various experimental studies, there are reports of 58 proteins interacting with ERRα. These interactions highlight the complex regulatory mechanisms and diverse cellular functions in which ERRα is involved in different tissues [32]. Among these manifold protein–protein interactions, the one with PGC-1α is singled out in the following paragraph for its functional relevance, which was recognized 20 years ago [36] and has since been confirmed in several tissues and physio-pathological conditions. PGC-1α operates at two distinct levels in its interaction with ERRα. Firstly, it induces the expression of ERRα, and secondly, it forms a complex with ERRα, facilitating the transcriptional activation of target genes. The expression of PGC-1α is known to be regulated in a tissue-selective manner by physiological signals that convey increased metabolic needs, such as cold exposure, physical exercise, or fasting [36,37,38,39,40,41], and PGC-1α orchestrates the transcription of genes relevant to mitochondrial biogenesis and oxidative metabolism, thus increasing energy production [39,42,43]. The fact that PGC-1α induces ERRα mRNA levels provides a molecular explanation for the high ERRα expression observed in tissues such as the heart, kidney, muscle, and brown fat that also highly express PGC-1α. Indeed, fasting, a condition known to induce PGC-1α expression in the liver, also increases ERRα mRNA levels [44]. Direct and reciprocal control on each other’s transcription is mediated by the presence of PGC-1α-specific binding sites on the promoter region of ERRα and ERRα-specific binding sites on the promoter of PGC-1α [45,46]. In the absence of PGC-1α, ERRα is a weak activator of transcription on its own. However, the co-expression of PGC-1α transforms ERRα into a potent transcriptional activator. These findings suggest that ERRα is not constitutively active and that its activation requires binding to PGC-1α. Surprisingly, this interaction differs from that of PGC-1α with other nuclear receptors. While PGC-1α typically recognizes most tested receptors (e.g., GR, glucocorticoid receptor; ER, estrogen receptor; TR, thyroid hormone receptor; RXR, retinoid X receptor; RAR, retinoic acid receptor; PPAR, peroxisome proliferator-activated receptor; HNF4, hepatocyte nuclear factor 4) via the canonical LXXLL motif L2, it can interact with ERRα equally well via the L2 or the L3 site. This differential utilization of Leu-rich motifs can serve to dissect the receptors involved in specific PGC-1α functions. For instance, L2A mutations disrupt GR-dependent but not ERRα-dependent effects of PGC-1 α, highlighting the potential use of L2 and L3 mutants of PGC-1α as valuable tools for unraveling the receptors recruiting PGC-1α at distinct promoters.

### 1.4. AMPK and SIRT1 Control the ERRα/PGC-1α Axis

It could be anticipated that the metabolic sensor AMP-activated protein kinase (AMPK) and NAD+-dependent deacetylase SIRT1 would be able to influence the activity of PGC-1α and ERRα [47,48]. Although the molecular consequences of these post-translational modifications are not yet fully understood, recent findings from various in vivo models strongly suggest that AMPK, SIRT1, PGC-1α, and ERRα form a functional axis to adapt energy metabolism to nutrient availability and energy expenditure at a tissue and organismic level. Chronic activation of AMPK results in the transformation of white into (thermogenic) beige adipose tissue via increased activity of the PGC-1α/ERRα axis. At an organismic level, this metabolic reprogramming of the adipose tissue results in increased resistance of mice to cold stress and reduced weight gain following a high-fat diet [49]. Caloric restriction, a means to activate the SIRT1/AMPK/PGC-1α axis in (cardio)myocytes [50], increases cardiomyocyte resistance to ischemic/reperfusion injury via enhanced mitochondrial energy metabolism [51]. In skeletal myocytes, the SIRT1/AMPK/PGC-1α/ERRα axis controls the transcriptional program that enables the switch of fiber type from glycolytic (fast twitch) to oxidative (slow twitch) [52]. Malfunctioning of the SIRT1/AMPK/PGC-1α/ERRα axis causes cardiomyopathy due to impaired mitochondrial energy metabolism [53]; conversely, its pharmacologic activation improves cardiac function via increased mitochondrial performance in a murine model of diabetic cardiomyopathy [50]. Interestingly, an antidiabetic drug used to inhibit tubular glucose uptake, but also unexpectedly inducing body weight loss, was found to activate the SIRT1/AMPK/PGC-1α axis in adipocytes, resulting in extensive remodeling of the adipose tissue (increased mitochondrial oxidative phosphorylation, fatty acid oxidation, and thermogenesis), likely the cause of the observed effect on body weight [54].

### 1.5. ERRs and Protection from Oxidative Stress, the Yang of Oxphos

It is reasonable to assume that since ERRα and PGC-1α enact a transcriptional program leading to increased mitochondrial respiration and oxidative metabolism, they should also activate molecular mechanisms to ensure increased protection from reactive oxygen species (ROS), which are an essentially unavoidable by-product of oxphos activity. Excess ROS can also be generated under conditions of hypoxia followed by re-oxygenation, a condition contributing to cell injury, particularly in eminently aerobic tissues, such as the heart, brain, and kidney. Addressing gaps in our understanding of the molecular protection mechanisms against ROS-derived tissue damage holds the potential for identifying new therapeutic strategies. In a recent study, ERRα and ERRγ are shown to both target an ROS-controlling transcriptional program, with different roles regarding glutamine fate, such as ERRα stimulating glutamine entry into the TCA (for energy production) and ERRγ instead favoring glutamine utilization to produce antioxidant glutathione. Since an elevated antioxidant capacity is a defining trait of breast cancers with poor treatment responses, the authors demonstrate that pharmacological inhibition of ERRα/γ enhances the antitumor efficacy of the chemotherapeutic paclitaxel on tumor organoids [55]. The overexpression of PGC-1α induces the expression of the key transcription factors controlling antioxidant defense nuclear factor erythroid 2-related factor 2 (NRF2) and hypoxia-inducible factor 1-α (HIF1α) [56], and ERRα induces the expression of genes involved in muscle cells and astrocyte response to hypoxia with HIF-1α-dependent and -independent mechanisms [57]. Another study, conducted in 3T3-L1 adipocytes, revealed that inhibiting ERRα activity using its inverse agonist XCT-790 significantly elevated ROS production [58]. Additionally, ERRα deficiency was found to worsen cisplatin-induced renal dysfunction, tubular injury, and oxidative stress [59]. 

### 1.6. The ABA-Sensing System Is an Ancient Stress-Sensing and -Responding Hormone/Receptors System

To endure environmental stress conditions that cannot avoid being sessile, land plants have evolved a range of responses, including growth inhibition, osmotic regulation, cuticular wax accumulation, leaf senescence, abscission, and dormancy controlled by several hormonal signals. Among these hormones, abscisic acid (ABA) serves a crucial function in transducing signals related to water, light, nutrient, osmotic, and oxidative stress conditions (see [60] for a recent review on the subject). ABA-triggered responses require hormone perception by specific cellular receptors, signal transduction mechanisms involving protein kinases and second messengers, the regulation of ion channels, and the activation of transcription factors controlling ABA-inducible genes (recently reviewed in [61]). ABA has an ancient evolutionary origin and is present and active in bryophytes, the first plants to conquer land, such ascyanobacteria, fungi, lichens, algae, and early Metazoa, such as sponges and hydroids [62,63], and also mammals [64]. In mammals, perhaps the most clinically relevant role of ABA is as an insulin-independent signal of nutrient (glucose) availability. An increase in blood glucose induces an increase in plasma ABA [65], and ABA stimulates insulin-independent muscle and adipocyte glucose uptake via GLUT4 and GLUT1 and increases mitochondrial respiration and oxidative metabolism in skeletal and cardiac muscle cells [66,67] and in brown and beige adipocytes [68]. Two mammalian ABA receptors have been identified so far, LANCL1 and LANCL2, belonging to a three-member protein family with a limited homology with bacterial lanthionine synthetase but lacking the ability to synthesize (antibacterial) lanthionines. Recombinant LANCL1 and LANCL2 bind ABA [66,69], and their overexpression increases while their double silencing conversely abrogates ABA sensing in mammalian cells [70]. Recent studies have shown that LANCL1 and LANCL2, whose genes likely evolved by gene duplication and are located on different chromosomes, have largely overlapping functions in eliciting ABA responses in myocytes and adipocytes; in view of their role in stimulating cell glucose uptake and mitochondrial function, receptor redundancy might serve as a protective mechanism to ensure ABA responsiveness in cases of receptor mutation or genetic loss. Interestingly, the genetic ablation of one of the two receptors results in the spontaneous overexpression of the other, both in vitro and in vivo [66]. LANCL3, the third member of the LANCL protein family, and possibly the product of a pseudogene, is expressed at a much lower level compared to LANCL1 and LANCL2. LANCL1 and LANCL2 are ubiquitously expressed in mammalian tissues, the highest levels being observed in the CNS, cardiomyocytes, and germinal cells, suggesting an important role in species and individual survival. Interestingly, triple LANCL knock-out (KO) mice die prematurely, for reasons that remain to be identified [71]. Recent results indicate that LANCL1/2 protein levels per se, even in the absence of ABA, control the transcription of the key players in the regulation of mitochondrial energy production, i.e., the metabolic sensors AMPK and SIRT1 and the transcriptional coactivators ERRα and PGC-1α. In cardiomyocytes, LANCL1/2 overexpression significantly stimulates, while their double silencing conversely severely reduces, mitochondrial number, respiration, proton gradient (despite an increased proton “leak”), cell duplication, contractile protein synthesis, ion channel transcription, and oxidative stress protection, improving cell survival to hypoxia [72]. Treatment with ABA further stimulates the functional features observed in the overexpressing but not double-silenced cells. Similarly to cardiomyocytes, LANCL1/2 overexpression in skeletal myocytes [66] and in brown/beige adipocytes [68] increases glucose uptake and oxidative metabolism, mitochondrial number, and proton gradient dissipation.

## 2. Cell- and Organ-Specific Functions of ERRα

### 2.1. Heart

In the heart, ERRα and ERRγ are essential for fetal cardiomyocyte development. The profound metabolic and structural modifications that fetal cardiomyocytes undergo to transform into contractile adult cells are indeed orchestrated by these transcription factors. Heart-specific KO of ERRα and ERRγ in mice demonstrates that they are essential not only for cardiomyocyte mitochondrial function but also for the activation of a complex gene transcription program impacting myocyte calcium signaling, contractile and cytoskeletal proteins, and ion channels. In addition, in the whole heart, ERRα and ERRγ reduce the expansion of fibroblasts, a hallmark of an ailing heart [73]. In vitro studies performed on human cardiomyocytes obtained from induced pluripotent stem cells confirm the fundamental role of ERRα in activating the master regulator PGC-1α, promoting mitochondrial biogenesis, respiration, and oxidative metabolism, which are all key characteristics of the mature cardiomyocyte [74]. The ERRα/PGC-1α axis indeed emerges as a pivotal regulator of mitochondrial energy metabolism in cardiomyocytes and is an eminently aerobic cell type, and recent studies are trying to define other molecular components of this transcriptional system. A new coactivator of the ERRα/PGC-1α axis was identified in the transcription factor PERM1. PERM1 expression levels increase during normal murine cardiomyocyte development and are conversely reduced in the human ailing heart, and its overexpression in murine neonatal cardiomyocytes increases the mitochondrial number, respiration, oxidative metabolism, and resistance to hypoxia/reoxygenation [75]. PERM1 physically interacts with both PGC-1α and ERRα and PERM1 KO mice show profound contractile, metabolic, and energy metabolism shortcomings [76]. These results suggest that pharmacologic activation of the PGC-1α/ERRα/PERM1 axis could improve cardiac function in different pathological conditions of the heart, including contractile dysfunctions caused by chronic heart diseases, mitochondrial dysfunction induced by doxorubicin therapy and hypoxia/reoxygenation-derived injury. Indeed, two different pan-ERR agonists were effective in protecting the murine heart in an in vivo model of heart failure, improving cardiomyocyte energy metabolism and contractility [77].

### 2.2. Skeletal Muscle

In type I fibers of skeletal muscle, mitochondrial number and metabolism are equally essential for cell function, as in the heart. Several studies have investigated the hallmarks of aging or ailing muscles, identifying a decline of ERRα/PGC-1α expression or function as a common feature. By studying the gene expression profiles of murine skeletal muscle during aging, Kan et al. identified a reduction in ERRα transcription and a concomitant loss of mitochondrial mass and metabolic proficiency as key features of the aging muscle [78]. More recently, the double KO of ERRα and ERRγ in mice was shown to lead to dramatic changes in muscle physiology, with a severe reduction in gene expression related to mitochondrial oxidative metabolism (and consequently energy production), histological abnormalities reminiscent of congenital myopathies, and marked exercise intolerance [79,80]. Only the double KO showed a severe phenotype, indicating functional redundancy of ERRα and ERRγ in the skeletal muscle [80]. Indeed, the overexpression of ERRγ alone is sufficient to transform type II into type I fibers, increasing mitochondrial oxidative metabolism (particularly fatty acid oxidation) and muscle strength in mice by activating the metabolic sensor AMPK [81], which, in turn, can activate PGC-1α [47,82]. Based on these studies, it is reasonable to hypothesize that ERRα/γ transcriptional activation should ameliorate muscle contractility in congenital myopathies and prevent the decline of muscle strength (dynapenia) and mass (sarcopenia) during aging. Exercise has been shown to upregulate the transcription of the PGC-1α/ERRα system in aging mice [83], providing a molecular basis for the long-known beneficial effect of mild but regular physical activity on muscle tonicity in the elderly. Several studies have attempted to achieve ERR activation by means of natural or synthetic compounds. Dietary intake of γ-Oryzanol, a mixture of ferulic acid esters of terpenoids similar to ABA, used as a sports supplement, was shown to improve muscle strength and exercise endurance in mice by upregulating the transcription of both ERRα/γ via PPARδ [84]. In another study, a synthetic pan-ERR agonist was shown to induce a transcriptional reprogramming of type IIA muscle fibers as observed in aerobic training and to increase physical endurance in mice [85]. Finally, a recently published study describes how the expression of a mutated form of ERRα, resistant to phosphorylation-induced inactivation, induces muscle fiber aerobic transformation and increases mitochondrial biogenesis, metabolic fuel oxidation, and physical endurance in mice [86].

### 2.3. Kidney

Mitochondria are the power stations of the cell; thus, it is not surprising that their dysfunction may have tissue-specific pathological consequences not only on cardiomyocyte performance but also on other eminently aerobic cell types. In the kidney, tubular cells require large amounts of metabolic energy for their re-absorptive and secretory activities. Moreover, their metabolism is eminently aerobic, and hypoxia, even if transient, can result in temporary or permanent loss of function. Similarly to what is observed in skeletal muscle and the heart, the expression of ERRα and PGC-1α is reduced in the aging kidney [87,88]. Several reports have highlighted the protective role of ERRα activation in different conditions of kidney damage. Treatment of mice with a pan-ERR agonist reversed the functional signs of age-related renal dysfunction, e.g., decreased mitochondrial function, loss of viable podocytes, increased albuminuria, and inflammation [88]. Using an opposite experimental approach, other authors showed that the genetic ablation of ERRα exacerbates the functional and ultrastructural kidney damage induced by cisplatin [59]. A direct relationship between (proximal) tubular cell differentiation, energy metabolism and function, and ERRα was established by means of single-cell transcriptomic analysis [89]. Although the scientific literature linking ERRα dysfunction to kidney disease is still limited, these reports indicate that further studies are warranted in a field of pivotal clinical interest given the high prevalence of chronic kidney disease (>10% of the population) worldwide, which is also related to the increase in diabetes cases.

### 2.4. Brain 

Similarly to what occurs in non-neuronal tissues, ERRα and PGC-1α cooperate in activating multiple transcriptional pathways in different regions of the brain, such as the cerebellum, hippocampus, and neocortex. Transcriptomic research on cell-specific functions of these transcriptional regulators has been hampered by the heterogeneity of neuronal populations present in these tissues. However, recent studies are shedding light on the role of the ERRα/PGC-1α axis in parvalbumin-positive interneurons (PV-INs). PV-INs are GABAergic interneurons that represent a small subpopulation of neurons present in different parts of the brain (cortex, hippocampus, and striatum) that control excitatory and inhibitory circuits and effectively fine-tune the firing activity of other neurons. A dysfunction of PV-INs is now believed to be involved in several neurological disorders characterized by cognitive and behavioral deficits, including schizophrenia, Alzheimer’s disease (AD), autism, and reward-seeking behavioral disorders [90]. PV-INs are fast-firing and high-energy demanding neurons and, as such, they are particularly rich in mitochondria and they express high levels of both PGC-1α and ERRα, the latter being necessary for the transcriptional program of PGC-1α [91]. Indeed, several behavioral abnormalities have been observed by different research groups in ERRα KO mice, including deficits in sensory gating, compulsive behaviors, and decreased sociability, which could be attributed to a dysfunction of inhibitory signals by interneurons and are reminiscent of human schizophrenia.

Interestingly, transcription and protein expression of ERRα were found to be age-dependently reduced in the cerebral cortex and hippocampus in a murine model of AD. In addition, in an in vitro model of human full-length β-amyloid precursor protein (APP) expression, the overexpression of ERRα was shown to reduce the molecular hallmarks of AD, i.e., processing APP into its amyloidogenic fragments and Tau phosphorylation [92]. In another study, expression levels of ERRα in the brain were shown to increase upon caloric restriction, and the silencing of ERRα in the prefrontal cortex of female mice results in an eating disorder reminiscent of anorexia [93], suggesting that ERRα dysfunction may be involved in the generation of eating disorders. Clearly, the study of the neurobiological functions of ERRα has just begun amid experimental hurdles imposed by the complexity of the brain, but it holds promise to unveil new pathogenetic mechanisms in a variety of neurological disorders still lacking effective therapy.

### 2.5. Brown Adipose Tissue (BAT)

White adipose tissue (WAT) is the repository of triglycerides, synthesized and hydrolyzed under the control mainly of the pancreatic hormones’ insulin and glucagon for whole-body fatty-acid-derived energy metabolism. Brown adipose tissue (BAT) is instead devoted to the thermal conversion of metabolic energy through the dissipation of the mitochondrial proton gradient by means of a specific protein “uncoupler” of oxphos, i.e., UCP-1. In adult humans, BAT is much less abundant than WAT, even in lean subjects; it is localized primarily around the major arteries in the chest and abdomen, and thermogenesis is controlled by thyroid hormones and sympathetic innervation. Brown adipocytes are particularly rich in mitochondria; thus, it could be anticipated that the PGC-1α/ERRα axis should be particularly relevant for their function. ERRα and ERRγ control the expression of several genes during white and brown adipocyte differentiation [94,95]. Murine primary brown adipocytes lacking all three ERR transcription factors (α, β, and γ), but not those lacking just one of the three, show a dramatic reduction in mitochondrial mass and oxidative capacity and also a hampered response to β-adrenergic stimulation [96]. In a subsequent study, it was shown that ablation of ERRα and ERRγ is sufficient to reproduce this phenotype [97], indicating that ERRβ is not essential. While this study highlights a redundancy of function by ERRα and ERRγ in brown adipocytes, another study has indicated a prominent role for ERRα in controlling the transcription of UCP-1, arguably the hallmark protein of BAT [98]. A clear indication of the critical role of the PGC-1α/ERRα axis in brown adipocyte function comes from a study employing mice with the adipose tissue-specific ablation of folliculin (FLCN), a strong repressor of AMPK. Constitutively active AMPK, in turn, activates the PGC-1α/ERRα axis in adipocytes, leading to a reprogramming toward a brown phenotype, with increased mitochondrial mass and oxidative metabolism, transcription of uncoupling proteins, increased energy expenditure and thermogenesis in the whole animal, and resistance to high-fat diet-induced weight increase [49]. In brown adipocytes, the histone deacetylase HDAC3 is a necessary coactivator of PGC-1α and ERRα to allow the transcription of UCP-1 [99]. Pharmacological activation of the AMPK/PGC-1α/ERRα axis has been proposed as a means to induce “beige” features in white adipocytes, which are far more abundant than brown adipocytes in humans, with the aim of increasing WAT energy dissipation to reduce body weight [100]. In view of the global trend toward an increase in the prevalence of overweight/obese subjects in developed as well as developing countries [101], pharmacological or nutraceutical means to induce the browning of WAT in humans are eagerly investigated [102,103] in view of the beneficial systemic metabolic effects observed in animal models of diabetes mellitus and fat liver disease upon increased energy expenditure [54,104]. 

### 2.6. ERRα in Tumors

Tumor cells require high amounts of metabolic energy not because of any cell-specific ATP-consuming function, as occurs in the heart, skeletal muscle, or neurons, but simply to divide. Although tumor cells can adapt to the low energy yield of glycolysis under hypoxic conditions, angiogenesis, often sustained autocrinally by tumor cells, allows mitochondrial oxidative metabolism, providing cells with higher amounts of metabolic energy [105]. Thus, it is not surprising that the transcriptional axis PGC-1α/ERRα is upregulated in several cancer cell types and that its inhibition could undermine cell energy production and thus proliferation. Pharmacologic inhibition or genetic ablation of ERRα reduced the proliferation of acute myeloid leukemia cells [106] and glioma cells [107], both in vitro and in vivo. The inhibition of ERRα function, in addition to reducing mitochondrial energy production, also undermines resistance of tumor cells to oxidative stress, by reducing NADPH content as a consequence of impaired glucose/glutamine oxidation [108], and to hypoxia [57]. Indeed, by controlling the transcription of enzymes and transporters crucial to energy metabolism, ERRα is also a fundamental contributor to the cell and tissue response to hypoxia. The activation of ERRα by tumor cells in response to radiation or chemotherapy provides a means to escape apoptosis; thus, ERRα inhibition has been recently proposed in combination with conventional antineoplastic therapy to maximize tumor cell death in esophageal [109] and in breast cancer [110]. In estrogen receptor-negative breast cancer cells, ERRα acts as an activating transcription factor. Its overexpression accelerates the proliferation of cancer cells in the mammary gland and upregulates the expression of vascular endothelial growth factor (VEGF), inducing neo-vascularization [20,111]. ERRα also increases the ability of breast cancer cells to utilize lactic acid as a metabolic substrate [112]. ERRα expression is also upregulated in urinary bladder carcinoma. The inhibition of ERRα expression leads to the inhibition of growth, proliferation, invasion, and migration of bladder carcinoma cells. This inhibition promotes cancer cell apoptosis and suppresses the epithelial–mesenchymal transition (EMT) of tumor cells [113]. In addition to its role as a metabolic master regulator, ERRα also stimulates cell migration by regulating different signaling pathways controlling actin filaments dynamics and focal adhesion assembly [114]. As cancer cell migration/adhesion properties are relevant to the process of metastasis generation, the inhibition of ERRα could provide an additional beneficial effect beyond metabolic castration.

## 3. The ABA/LANCL1-2 Hormone/Receptor System Controls the PGC-1α/ERRα Axis 

Interestingly, the tissues and organs with high expression levels of ERRα (type I skeletal muscle fibers, cardiomyocytes, brown adipocytes, and neuronal cells) also show high levels of transcription of LANCL1 (https://www.proteinatlas.org/ENSG00000115365-LANCL1 accessed on 23 April 2024) and LANCL2 (https://www.proteinatlas.org/ENSG00000132434-LANCL2 accessed on 23 April 2024). The co-expression of the PGC-1α/ERRα axis and LANCL1/2 may have suggested their possible functional collaboration in view of the fact that skeletal myocytes overexpressing LANCL1/2 show the typical functional features activated by ERRα, such as increased mitochondrial number and enhanced mitochondrial respiration via the activation of the AMPK/PGC-1α pathway [66], which, in turn, activates ERRα function and transcription with a feed-forward relay. The signaling cascade triggered by ABA via the LANCL1/2 proteins in skeletal muscle indeed involves the AMPK/PGC-1α/SIRT1 axis, linked to ERRα-mediated transcriptional effects via AMPK and PGC-1α, leading to increased gene transcription and protein overexpression of glucose transporters GLUT1 and GLUT4, the NAD-synthesizing enzyme Nampt, the RabGAP TBC1D1, and muscle-specific mitochondrial uncoupling proteins UCP-3 and sarcolipin, along with an augmented mitochondrial DNA content and respiration [66]. These ABA-induced transcriptional and translational effects contribute to increased glucose uptake and energy metabolism in the muscle, resulting in elevated muscle glucose consumption in vivo [67]. Additionally, LANCL1 overexpressing and LANCL2 KO mice exhibit higher skeletal muscle mitochondrial DNA content and increased expression levels of AMPK, PGC-1α, GLUT1/4, Nampt, and UCP-3 compared to wild-type (WT) mice, levels that further increase after ABA administration. Although a role for ERRα in mediating these effects of the ABA/LANCL1-2 system in skeletal muscle was not explored in this study, it is likely that future studies will confirm a role for ERRα in the signaling pathway downstream of LANCL1/2 in skeletal myocytes, as observed in cardiomyocytes (Section 3.1). 

### 3.1. Cardiomyocytes

In cardiac myocytes, mitochondrial function and biogenesis are regulated by the PGC-1α/ERRα transcriptional coactivator team (Section 2.1). Previous studies have shown that the ABA/LANCL1-2 system controls mitochondrial function in skeletal myocytes, increasing mitochondrial DNA content and respiration through a signaling pathway involving AMPK, PGC-1α, and SIRT1, and ERRα is required for the transcriptional effects of PGC-1α on genes critical for mitochondrial energy production in cardiac and skeletal muscle [72]. In this recent study, aimed at unraveling the relationship between the ABA/LANCL1-2 system, ERRα, and mitochondrial function in cardiomyocytes, the overexpression of LANCL1/2 significantly enhanced and the double silencing conversely reduced several key functional features of rat H9c2 cardiomyocytes. The observed effects of LANCL1/2 overexpression included i) increased mitochondrial respiration, with higher basal and maximal respiration rates, and the doubling of the spare respiratory capacity and a steeper proton gradient (ΔΨ); ii) enhanced the fatty acid-fueled respiration rate; iii) reduced mitochondrial ROS content and led to higher expression levels of ROS-scavenging enzymes and lower levels of ROS-producing enzymes; iv) increased the transcription and expression of contractile and ion channel proteins; v) improved resistance to hypoxia/reoxygenation; and vi) increased the proliferation rate. In essence, all functional features of LANCL1/2-overexpressing cells are consistent with the activation of the AMPK/PGC-1α/SIRT1 signaling pathway. The additional information provided in this study is the identification of ERRα as a key player since ERRα silencing abrogates all transcriptional and functional effects observed in LANCL1/2-overexpressing cells. Moreover, ERRα appears to be linked to the levels of LANCL1/2 expression through a reciprocal feed-forward mechanism of transcriptional stimulation. LANCL1/2 overexpression increases ERRα transcription and expression, while silencing ERRα reduces endogenous LANCL1/2 mRNA levels. This study also highlights the direct role of ERRα in the regulation of the cell cycle. Several cell cycle-controlling genes, such as CCNDs and E2Fs, are upregulated in LANCL1/2 overexpressing and downregulated in double-silenced H9c2 cells. Silencing of ERRα negatively affects the transcriptional levels of these cyclins and impairs cardiomyocyte proliferation, indicating a role for ERRα in the growth-promoting effect of the ABA/LANCL system. These results suggest that the ABA/LANCL1-2/ERRα system acts as a new regulator of the complex gene network involved in cell cycle progression and energy metabolism in cardiomyoblasts. Cyclins and CDKs, known for their roles in cell cycle progression, also play crucial roles in the regulation of energy metabolism, as cell division requires high amounts of ATP. The close interconnection between nutrient availability, metabolic energy production, mitochondrial activity, and cell division requires coordinated regulation. This study indicates that LANCL1/2 overexpression increases the transcription of cell cycle- and metabolism-controlling cyclins through ERRα, while their silencing has the opposite effect, warranting further investigation, especially in the context of heart pathological conditions where pharmacological stimulation of this system may have therapeutic potential. In addition to transforming cardiomyoblasts into a sort of “super-performing” cells, this study also unveils a protective effect of LANCL1/2 overexpression on oxidative stress. Despite an increased mitochondrial respiratory activity, which is expected to generate ROS, LANCL1/2-overexpressing cells exhibit reduced mitochondrial ROS production. The reduced ROS generation in LANCL1/2-overexpressing cells may be attributed, at least in part, to the elevated expression of ROS-scavenging enzymes in these cells. Silencing ERRα in LANCL1/2-overexpressing cells results in a significant increase in ROS production, indicating that ERRα is necessary for mediating the protective effects of LANCL1/2 on ROS generation. These findings thus identify the ABA/LANCL1-2/system as an unrecognized controller of ERRα, and thus of mitochondrial energy metabolism, ROS management, and cell proliferation in H9c2 cardiomyoblasts. The study by Spinelli et al. also explores the relationship between a mild proton leak across the inner mitochondrial membrane, respiratory chain function, and ROS generation in the context of LANCL1/2 overexpression. A proton leak, considered a means to enhance respiratory chain function and reduce ROS generation, is typically associated with a reduction in the proton motive force (ΔΨ) required for ATP synthesis. However, the results indicate a higher proton leak in the face of an increased ATP-dependent respiration in LANCL1/2-overexpressing cells, challenging the conventional understanding. These results suggest that a mild proton leak, occurring through ANT1 and UCP-3, whose transcription is upregulated in overexpressing cells, may in fact maximize ATP production. By facilitating electron transport through the reduction in the ΔG for proton pumping at respiratory complexes I and III, a mild proton leak allows the orderly flux of electrons to oxygen and prevents their retrograde flux, which can otherwise occur at these complexes, resulting in ROS generation [115,116,117]. Further investigations are needed to understand the molecular mechanisms underlying the “fine-tuning” of oxphos observed in LANCL1/2-overexpressing cells via ERRα. 

### 3.2. Adipocytes

At both the transcriptional and protein levels, LANCL proteins play a crucial role in upregulating energy metabolism and mitochondrial respiration in brown and "beige" white adipocytes, contributing to their energy-dissipating and thermogenic functions. Direct transcriptional control by the ABA/LANCL system on ERRα in adipocytes is demonstrated by the observation that the overexpression of LANCL1/2 increases 20-fold, while their combined silencing dramatically reduces ERRα mRNA levels in both white and brown adipocytes. In addition, ABA simulates ERRα expression in the BAT of WT mice, and ERRα is spontaneously overexpressed in the BAT of LANCL1-overexpressing, LANCL2 KO mice. The metabolic and functional consequences of the activation of the LANCL1-2/PGC-1α/ERRα axis in white and brown adipocytes, triggered by the overexpression of the LANCL proteins, and further augmented by ABA, can be summarized as follows: i) enhanced glucose transport (via GLUT4 upregulation) and oxidation, mitochondrial biogenesis (MT-DNA), respiration (increased expression of complex I and of basal and maximal O2 consumption), and ∆Ψ magnitude; ii) upregulation of the expression of receptors for browning hormones (ADRβ3, THRα1/β), the enzyme deiodinase (converting T4 into the active T3), and uncoupling proteins (UCP-1/3); iii) 2- and 4-fold increase in mitochondrial DNA and oxphos complex I (MT-ND1), respectively, in the BAT from ABA-treated mice; iv) upregulation of receptors for "browning" hormones in human ABA-treated adipocytes and in the BAT from ABA-treated mice; v) a higher expression of MT-ND1, thyroid hormone receptors, and ERRα in LANCL1-overexpressing LANCL2 KO mice compared to WT mice. In conclusion, this study provides evidence of transcriptional and functional cooperation between the LANCL1/2 proteins and ERRα in brown adipocytes, at the same time indicating potential therapeutic strategies for modulating brown adipocyte metabolism and function [68].

## 4. Conclusions and Future Perspectives

### 4.1. Conclusions

The principal conclusions that can be drawn from the literature discussed in this review can be summarized as follows: Enhancing ERRα activity promotes mitochondrial health and protects against oxidative stress in several aerobic organs and tissues, particularly in the heart, skeletal muscle, brain, and kidney.Repressing ERRα activity is a means to undermine mitochondrial energy metabolism (supporting cell duplication) and cytoskeletal dynamics (allowing cell migration) in cancer cells.The ABA/LANCL1-2 hormone/receptor system emerges as a new controller of ERRα expression levels and transcriptional activity via the AMPK/SIRT1/PGC-1α axis.A reciprocal feed-forward transcriptional relationship exists between the LANCL proteins and transcriptional coactivators ERRα/PGC-1α, which could be exploited with natural or synthetic LANCL agonists [118] to improve mitochondrial function in multiple clinical settings.

### 4.2. Future Perspectives

The recent appreciation of the role of the ABA/LANCLs hormone/receptor system in controlling the ERRα/PGC-1α transcriptional complex and in fostering its genome-wide effects provides new molecular tools to activate these pivotal master regulators of cell energy metabolism and, in general, improve cell “fitness” to its specific functional role. Targeting the LANCL/PGC-1α/ERRα axis is expected to improve cell and tissue functions in several organs, leading to favorable whole-body consequences. A graphical summary is shown in Figure 2.

#### 4.2.1. Increase Muscle and Adipocyte Energy Expenditure 

By targeting the PGC-1α/ERRα transcriptional complex via LANCL1/2, ABA holds promise to achieve the stated aim with a natural (not pharmacological) endogenous agonist, which is also present in vegetal extracts. Indeed, targeting ERRα with pharmacologic agonists has been proposed as a new strategy to combat metabolic diseases [5]. Notably, ABA-treated LANCL1-overexpressing LANCL2 KO, mice show a significant increase in the transcription of key glycolytic enzymes (GaPDH, PFK1), PDH, and pyruvate and fatty acid transporters in skeletal muscles compared to untreated controls, stimulating muscle oxidative energy production. Interestingly, female mice with this genotype, fed a high-glucose diet for three months, display significantly lower body weight gain compared with WT siblings, despite increased food intake [119]. These findings suggest that ABA/LANCL-mediated mitochondrial uncoupling in muscles and adipocytes, together with increased oxygen consumption, may influence whole-body energy expenditure.

#### 4.2.2. Reduce Oxidative Stress

ERRα has been known for many years to mediate the transcription of several gene-encoding enzymes involved in protection from oxidative stress [120]. More recently, ERRα and ERRγ have been involved in protection from oxidative stress via glutamine utilization for glutathione synthesis [55]. Oxidative stress induced by cisplatin in the kidney is reduced by ERRα overexpression/activation [59]. ERRα also protects cardiomyocytes from oxidative injury resulting from hypoxia/reoxygenation [75]. In tumor cells, mechanisms protecting agsinst drug-induced oxidative stress may conversely be reduced by ERRα inverse agonists, leading to increased tumor sensitivity to chemotherapy [55]. As the ABA/LANCL system lies upstream of ERRα and controls its transcription and activity, it can be anticipated that ABA itself or synthetic LANCL agonists [118] should improve antioxidant defense in several cell types.

#### 4.2.3. Improve Cardiomyocyte Function

Targeting the PGC-1α/SIRT1/ERRα axis has already been considered a strategy to enhance cardiomyocyte function, particularly in conditions of chronic heart disease, such as diabetic cardiomyopathy. Identification of the ABA/LANCL1-2 system as an essential part of the ERRα activating pathway suggests that ABA could serve as a pharmacological agonist to modulate this pathway. Additionally, the LANCL proteins themselves emerge as potential molecular targets for intervention. The question of the molecular signal that activates endogenous LANCL1/2 transcription in cardiomyocytes remains open. Nitric oxide (NO) is a candidate signal, given its involvement in positive feedback linking endothelial nitric oxide synthase (eNOS) transcription and activity to LANCL1/2 and ERRα expression levels [72]. NO is known to be produced by the beating heart, and its levels can be influenced by conditions of cardiomyocyte stress. The reciprocal feed-forward mechanism observed in the activation of LANCL1/2, ERRα, and NO generation in cardiomyocytes highlights the complexity of the signaling network. It is noteworthy that LANCL1/2 overexpression and ABA treatment also lead to an increase in the transcription of the rate-limiting enzyme in the synthesis of tetrahydrobiopterin (BH4), a coenzyme crucial for preventing the "uncoupling" of NOS, ensuring NO instead of ROS generation. Furthermore, BH4 has been implicated in stimulating mitochondrial biogenesis and cardiac contractility through PGC-1α, suggesting a potential contribution to the coordinated improvement of mitochondrial performance observed in the LANCL/ERRα pathway. In short, exploring the use of ABA to modulate the LANCL1-2/ERRα signaling pathway and understanding the molecular signals involved in the activation of LANCL1/2 in cardiomyocytes could pave the way for new therapeutic strategies targeting mitochondrial function and cardiomyocyte fitness. The overexpression of LANCL1/2 in rat cardiomyoblasts increases contractile protein synthesis, mitochondrial number, respiration, and cell size [72]. These results warrant further studies to explore the possibility of a translation to the clinical setting.

#### 4.2.4. Improve Skeletal Muscle Fitness

Physical exercise physiologically upregulates the expression of PGC-1α/ERRα in muscles, which mediate the modifications of muscle metabolism and contractility, which we collectively call “fitness”, such as increased mitochondrial number, oxphos activity, metabolic flexibility, type I fiber content, and vascularization. However, there are clinical conditions where physical exercise is either not possible or recommended due to coexisting diseases. In these cases, molecular agonists capable of reproducing the benefits of exercise on the diseased or aging muscle may represent an alternative to muscle wasting. Recent reports highlight the usefulness of targeting the PGC-1α/ERRα axis to improve muscle fitness by means of synthetic or natural ERRα agonists [85,121] or ABA [67].

#### 4.2.5. Improve Whole-Body Glucose Disposal to Combat Diabetes 

By increasing muscle and adipose tissue glucose uptake by means of what we now recognize as the LANCL1/2-mediated activation of the AMPK/PGC-1α/ERRα axis, ABA improves whole-body glucose disposal with an insulin-independent mechanism [67]. In a protocol involving ABA pretreatment followed by diabetes induction with low-dose streptozotocin (STZ), LANCL2 KO mice exhibited significantly lower mean glycemia than WT animals, a result attributable to LANCL1 overexpression substituting for LANCL2 in ABA binding and pathway activation. In the same study, WT mice treated with ABA show an almost 10-fold increase in the transcription of the insulin receptor, indicating that prolonged ABA treatment may enhance muscle sensitivity to both endogenous and exogenous insulin [122]. Since the transcriptional effects of insulin itself depend on ERRα [123], targeting ERRα should improve insulin sensitivity. Indeed, most recently, a study described the favorable effects of a non-specific ERR agonist on an animal model of obesity and metabolic syndrome [124].

#### 4.2.6. Explore the Neuroprotective Role of the LANCL/ERRα System in the Brain

ABA is particularly abundant in the brain of mammals [125], and the highest level of expression of LANCLs in the body occurs in the brain, particularly in the cerebral cortex, spinal cord, and hippocampus (https://www.proteinatlas.org/ accessed on 23 April 2024). These are also among the brain regions with the highest transcriptional levels of ERRα, suggesting a functional cooperation between the ABA/LANCL system and ERRα, the nature and scope of which are open to investigation. In a cell model of amyloidogenesis, the overexpression of ERRα reduced amyloid accumulation and Tau phosphorylation, and in a mouse model of Alzheimer’s disease, ERRα mRNA and protein levels were found to be reduced in the brain [92]. ERRα activation has been proposed to improve mitochondrial function in the brain in neurodegenerative and traumatic/hypoxic injuries [126,127], and the ABA/LANCL system may provide the means to achieve its transcriptional and functional upregulation, another research area of significant scientific interest and potentially relevant clinical impact. Interestingly, the overexpression of LANCL1 was shown to improve mitochondrial function and preserve cell vitality in an in vitro model of neuronal death induced by glucose and oxygen deprivation via the activation of PGC-1α [128]. Although it was not investigated in this study, a possible role for ERRα is highly probable, in light of the more recent literature.

#### 4.2.7. Explore the Nature of the Molecular Interaction between the ABA/LANCL System and ERRα 

Finally, the recently unveiled functional interaction between the ABA/LANCL system and ERRα raises some interesting questions regarding their possible physical interaction, which will be addressed in future studies. Is ABA the natural ligand of ERRα? Indeed, the lipophilic nature of ABA, its small molecular weight, and its ancient evolutionary origin (preceding estrogens) suggest exploring this possibility. A precursor of mammalian ERRα is already present in marine invertebrates [129], which possess ABA [63], but estrogen presence and function are still debated [130]. Indeed, estrogen receptors appear to have evolved from ERRs [131]. Do LANCL proteins and ERRα physically interact? Interestingly, LANCL2, which is bound to the internal side of the plasmamembrane via a myristoyl anchor and interacts with a G-protein, undergoes nuclear translocation upon ABA binding, a rare (perhaps unique) example of a receptor combining the typical features of peptide and steroid receptors [132]. Thus, it is possible that LANCL2 may form a complex with PGC-1α/ERRα and participate in the transcriptional function of these transcription factors. 

## Figures and Tables

**Figure 1 ijms-25-04796-f001:**
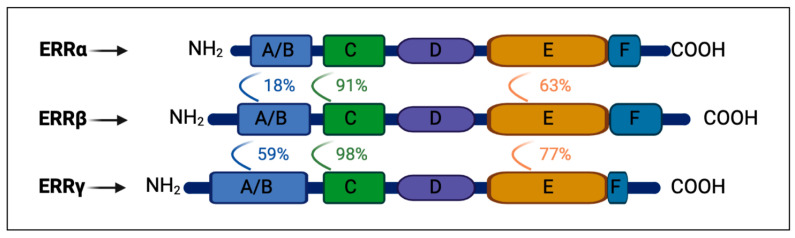
The protein architecture of the three estrogen-related receptors (ERRs). ERRs consist of six conserved regions (A/B, C, D, and E/F domains). The number between the two receptors indicates the sequence identity of the corresponding domain between different receptors.

**Figure 2 ijms-25-04796-f002:**
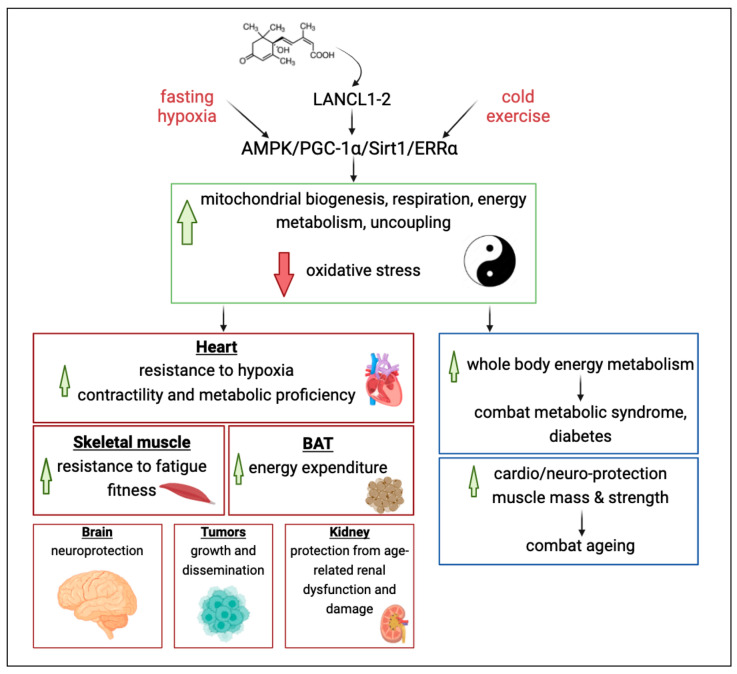
The LANCL/ERRα/PGC-1α axis: uncoupling the yin and yang of mitochondrial function. Several feed-forward mechanisms concur in maintaining the LANCL1-2/AMPK/SIRT1/ERRα/PGC-1α axis active once “started”. AMPK stimulates Nampt activity and NAD synthesis, consequently increasing SIRT1 activity; AMPK and SIRT1 post-translationally modify and activate PGC-1α; LANCL1/2 expression levels upregulate the expression of AMPK, ERRα, PGC-1α, and ERRα, which, in turn, control LANCL1/2 expression [68,70,72]. This signaling axis stimulates mitochondrial biogenesis and oxphos activity and upregulates the cell’s ability to cope with increased oxidative stress at the same time, which is linked to a higher respiration capacity. Environmental stresses, such as variations in nutrient, oxygen, temperature, or cell energy status, and the stress hormone ABA, trigger the activation of this highly conserved signaling axis. This axis, in turn, exerts transcriptional control over hundreds of genes involved in key cell processes in tissues and organs with high energy demands, resulting in increased mitochondrial energy production and antioxidant defense. These responses affect the whole-body energy balance; thus, agonists of this signaling axis could be used not only to improve organ-specific diseases but also to ameliorate systemic conditions with multi-organ failures, such as diabetes and aging.

## Data Availability

Not applicable.

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
