# Peer review of "Estrogen-Related Receptor α: A Key Transcription Factor in the Regulation of Energy Metabolism at an Organismic Level and a Target of the ABA/LANCL Hormone Receptor System"

_ijms, 2024, doi:10.3390/ijms25094796_

Round 1

Reviewer 1 Report

Comments and Suggestions for Authors

In this article, the authors review the topic of the ERRa as a key transcription factor in the regulation of energy metabolism at an organismic level. Moreover, they document a connection to the ABA/LANCL hormone receptor system for which the ERRa is a suggested target.

General comment:

The topic of this article is relevant. The article is well written and very informative, still, it is easy to follow. Although a bit unusual in regard to the length of the introductory and future perspectives part, the article is still well structured with a straightforward logical flow. In general, it would be useful to share this article with a scientific community.

Minor points:

Figure 2: Would it be possible to add the tumor role of ERRa on the figure? This is only a suggestion and I leave it completely up to authors to decide whether this would be suitable.

Lines 131-132, 'e.g. GR, ER, TR, RXR, RAR, PPAR, HNF4': The abbreviations (the protein names) could be explained.

Author Response

The Authors wish to thank the Reviewer for taking time to read the manuscript and solicit changes, which improved the quality of the paper.

The following is a point-by-point reply to criticisms.

Reviewer #1

In this article, the authors review the topic of the ERRa as a key transcription factor in the regulation of energy metabolism at an organismic level. Moreover, they document a connection to the ABA/LANCL hormone receptor system for which the ERRa is a suggested target.

General comment:

The topic of this article is relevant. The article is well written and very informative, still, it is easy to follow. Although a bit unusual in regard to the length of the introductory and future perspectives part, the article is still well structured with a straightforward logical flow. In general, it would be useful to share this article with a scientific community.

Minor points:

Figure 2: Would it be possible to add the tumor role of ERRa on the figure? This is only a suggestion and I leave it completely up to authors to decide whether this would be suitable.

As suggested by the Reviewer, the role of ERRα in tumors and also in the kidney has been added to Figure 2.

Lines 131-132, 'e.g. GR, ER, TR, RXR, RAR, PPAR, HNF4': The abbreviations (the protein names) could be explained.

The names of the proteins after the relative abbreviations have been inserted into the text of the manuscript.

Reviewer 2 Report

Comments and Suggestions for Authors

The article titled " ERRα, a key transcription factor in the regulation of energy metabolism at an organismic level and a target of the ABA/LANCL hormone receptor system" by Sonia Spinelli et al. provides valuable insights. However, there are several areas that require attention:

1. The authors should provide their own justification for the study, as previous publications have already explored the relevance of the topic. Examples of such publications include articles in PubMed, such as Antioxidants (Basel). 2023 Aug 30;12(9):1692. doi: 10.3390/antiox12091692; Int J Mol Sci. 2023 Jan 7;24(2):1199. doi: 10.3390/ijms24021199;  Preprint, 10.20944/preprints202308.0328.v1; among others. Consequently, the study does not provide any innovative information.

2. Abscisic acid (ABA) is usually acknowledged as a plant stress hormone but is also detected and functions in organisms beyond the plant kingdom. However, the authors did not provide specific evidence or literature of ABA's detection and function in non-plant organisms.

3. As stated on line 575-578, the ABA/LANCL system lies upstream of ERRα and controls its transcription and activity. It can be anticipated that ABA itself or synthetic LANCL agonists should improve antioxidant defense in several cell types. However, the authors did not provide specific evidence or literature related to synthetic LANCL agonists.

4. The authors could make a table or figures and provide cell- and organ-specific functions of ERRα and its key role in energy metabolism using in vitro cell line models and in vivo studies.

5. Under the subtitle "4. Conclusions and Future Perspectives," the authors provided numerous subtitles and data references. It would be better for the authors to provide concise conclusions and future perspectives. They can give all the given information into different subtitles.

Comments on the Quality of English Language

Moderate editing of English language required

Author Response

The Authors wish to thank the Reviewer for taking time to read the manuscript and solicit changes, which improved the quality of the paper.

The following is a point-by-point reply to criticisms.

Reviewer #2

The article titled " ERRα, a key transcription factor in the regulation of energy metabolism at an organismic level and a target of the ABA/LANCL hormone receptor system" by Sonia Spinelli et al. provides valuable insights. However, there are several areas that require attention:

1. The authors should provide their own justification for the study, as previous publications have already explored the relevance of the topic. Examples of such publications include articles in PubMed, such as Antioxidants (Basel). 2023 Aug 30;12(9):1692. doi: 10.3390/antiox12091692; Int J Mol Sci. 2023 Jan 7;24(2):1199. doi: 10.3390/ijms24021199;  Preprint, 10.20944/preprints202308.0328.v1; among others. Consequently, the study does not provide any innovative information.

The point is that the present manuscript is a Review, not an experimental Article, and as such it aims at summarizing recently published results on the topic ERRα and the ABA/LANCL system in order to provide the interested reader with an updated and comprehensive information on this evolving topic. The articles cited by the Reviewer, and also cited in this Review, are briefly summarized here and contribute to this information. The aim is not to “provide innovative information”, but to summarize the existing one, putting it into the context of the current state-of-the-art, as any Review aims to achieve.

2. Abscisic acid (ABA) is usually acknowledged as a plant stress hormone but is also detected and functions in organisms beyond the plant kingdom. However, the authors did not provide specific evidence or literature of ABA's detection and function in non-plant organisms.

The evidence of the detection and function of abscisic acid (ABA) in non-plant organisms has been described in Paragraph 1.6, although not in-depth, as there are other papers (doi: 10.1073/pnas.261448698, 10.1074/jbc.M405348200, 10.1007/s10265-011-0410-5, 10.1096/fj.11-190140, 10.1016/j.molmet.2021.101263, 10.1038/s41598-020-58206-0, 10.3390/ijms24043489), which focus specifically on this topic.

3. As stated on line 575-578, the ABA/LANCL system lies upstream of ERRα and controls its transcription and activity. It can be anticipated that ABA itself or synthetic LANCL agonists should improve antioxidant defense in several cell types. However, the authors did not provide specific evidence or literature related to synthetic LANCL agonists.

A research article cited in the Review regards the antioxidant effect of ABA in a hypoxia/reoxygenation model of cardiomyocyte injury (doi: 10.3390/antiox12091692). As ABA is the natural ligand of the LANCL proteins, this Review focuses on its role as a LANCL agonist. However, in the conclusions we mention the possibility that synthetic agonists could be developed in the future to mimic the effect of ABA. A recently published study on this subject has been added to the conclusions and future perspectives (doi: 10.3390/pharmaceutics15122754).

4. The authors could make a table or figures and provide cell- and organ-specific functions of ERRα and its key role in energy metabolism using in vitro cell line models and in vivo studies.

Figure 2 has been implemented with the role of ERRα also in tumors and kidney, including the main related functions.

5. Under the subtitle "4. Conclusions and Future Perspectives," the authors provided numerous subtitles and data references. It would be better for the authors to provide concise conclusions and future perspectives. They can give all the given information into different subtitles.

The subtitles are meant to aid the reader to appreciate the numerous and different future areas of promising investigation. In our view, removing the subtitles does not aid comprehension. A very concise summary is already provided in Figure 2.

Reviewer 3 Report

Comments and Suggestions for Authors

This is a review about ERRalpha, proposing a major role of this orphan nuclear receptor in regulation of energy metabolism. I have the following comments:

1. In the title ERRalpha is called 'transcription factor'. Is this correct and fitting its function as nuclear receptor? It might be better to use transcriptional regulator to avoid confusion (see also point 4 below).

2. In the abstract ABA is mentioned - a plant hormone which role in mammals has been only acknowledged in papers from the authors. I would suggest to rewrite the abstract with the main focus on ERRalpha and PGC-1alpha (as done in the review).

3. The part 'future perspectives' (paragraph 4.2 ff) is very long and highly speculative. I would strongly recommend to leave it out from the review and place Fig. 2 to the section 3.

4. Please use the term transcription factor only when its an established one. Even PGC-1alpha is correctly peroxisome proliferator-activated receptor γ coactivator 1-α - a transcriptional coactivator which promotes the activation of multiple transcription factors.

Author Response

The Authors wish to thank the Reviewer for taking time to read the manuscript and solicit changes, which improved the quality of the paper.

The following is a point-by-point reply to criticisms.

Reviewer #3

This is a review about ERRalpha, proposing a major role of this orphan nuclear receptor in regulation of energy metabolism. I have the following comments:

1. In the title ERRalpha is called 'transcription factor'. Is this correct and fitting its function as nuclear receptor? It might be better to use transcriptional regulator to avoid confusion (see also point 4 below).

ERRα exerts its transcriptional regulatory function by making a complex with other proteins, which act together to start gene transcription. ERRα binds to the estrogen-related receptor response element (ERRE) to regulate gene transcription (doi: 10.1007/s00044-019-02493-4); thus, in the end, it acts as a transcription factor. We feel that a very subtle semantic, but not biological difference, is expressed by the terms transcription factor or transcription regulator, when applied to ERRα.

2. In the abstract ABA is mentioned - a plant hormone which role in mammals has been only acknowledged in papers from the authors. I would suggest to rewrite the abstract with the main focus on ERRalpha and PGC-1alpha (as done in the review).

In fact, the new information provided in this Review, and discussed in the light of current knowledge regarding the role of the transcriptional regulators ERRα/PGC-1α, is exactly that the ABA/LANCL system emerges as a new player in the regulation of the ERRα/PGC-1α axis. This information is also summarized in the title. Rewriting the abstract omitting this fact does not convey the purpose of the Review.

3. The part 'future perspectives' (paragraph 4.2 ff) is very long and highly speculative. I would strongly recommend to leave it out from the review and place Fig. 2 to the section 3.

The future perspectives are indeed quite long; however, the Authors believe that they convey the sense of the multiple and exciting areas of investigation that this Review aims at emphasizing. A very succinct summary is already provided in Figure 2, but a somewhat more extensive description is in our view necessary to allow the reader to fully appreciate the clinical potentialities that research in this area may offer. Very synthetic bullet-point conclusions are provided, but they do not allow to fully appreciate these perspectives.

4. Please use the term transcription factor only when its an established one. Even PGC-1alpha is correctly peroxisome proliferator-activated receptor γ coactivator 1-α - a transcriptional coactivator which promotes the activation of multiple transcription factors.

The term "transcriptional coactivator" has been correctly placed alongside PGC-1α in the text of the manuscript.

Reviewer 4 Report

Comments and Suggestions for Authors

In this review article ERRα a key Transcription Factor in the Regulation of Energy Metabolism at an Organismic Level and a Target of the ABA/LANCL Hormone Receptor System. From the review as written by the authors is not clear the mechanisms through which ERRα regulate energy metabolism in the different organs. A major question is whether if their is organ specific differences in ERRα activity and energy metabolism. What are the benefits for these differences and potential physiological relevance for focusing on them. How will this advance the field. The authors should add table and/or figures showing specific organ targets by ERRα to regulate metabolisms.

Author Response

The Authors wish to thank the Reviewer for taking time to read the manuscript and solicit changes, which improved the quality of the paper.

The following is a point-by-point reply to criticisms.

Reviewer #4

In this review article ERRα a key Transcription Factor in the Regulation of Energy Metabolism at an Organismic Level and a Target of the ABA/LANCL Hormone Receptor System.

From the review as written by the authors is not clear the mechanisms through which ERRα regulate energy metabolism in the different organs.

The literature cited in this Review deals extensively with the different gene targets and functional effects induced by activation of the ERRα/PGC-1α transcriptional regulator complex. We provide a summary of these data, which involve, but are not limited to, an increase in mitochondrial number, respiration, proton gradient, oxidative metabolism (glucose and fatty acid transport and oxidation) and ultimately ATP production.

A major question is whether if their is organ specific differences in ERRα activity and energy metabolism.

Some peculiarities regarding specific organs and tissues are indicated in the various paragraphs, as reported in the literature.

What are the benefits for these differences and potential physiological relevance for focusing on them. How will this advance the field.

The authors believe that these issues have been addressed in the (quite lengthy) future perspectives.

The authors should add table and/or figures showing specific organ targets by ERRα to regulate metabolisms.

The point is that, by improving mitochondrial activity, the LANCL/ERRα/PGC-1α axis allows each organ or tissue described in this Review to perform better in its specific function. An increased mitochondrial energy production underlies all clinically relevant beneficial effects described in the literature as a consequence of activating this axis, as summarized in Figure 2. Although this conclusion may not come as a surprise, given that mitochondria are the power central of every cell type, it is noteworthy that a single signaling axis can positively regulate cell energy production through a complex network of gene targets, as detailed in the cited literature. Figure 2 has been implemented with the role of ERRα also in tumors and kidney, including the main related functions.

Round 2

Reviewer 2 Report

Comments and Suggestions for Authors

Accept in present form

Comments on the Quality of English Language

Moderate editing of English language required

Author Response

The text of the manuscript has been reviewed by a native English-speaking colleague, who expressed appreciation for the language and did not find any need for improvement.

Reviewer 3 Report

Comments and Suggestions for Authors

The authors have addressed most of my comments.

Author Response

We thank the Reviewer for the suggested good comments that improve the manuscript.

Reviewer 4 Report

Comments and Suggestions for Authors

The authors did not address the concerns that I raised in my first review of their manuscript.

Author Response

We are deeply concerned about the continuing concerns of the Reviewer, but we tried our best to understand the criticisms raised and to reply to them; despite our effort it seems that we did not succeed.

Regarding the first issue “From the review as written by the authors is not clear the mechanisms through which ERRα regulate energy metabolism in the different organs”, we cite several published articles focused on the targets of ERRα activity, which enable its stimulatory effect on mitochondrial energy metabolism in different tissues and organs (refs 23, 25-28). A comprehensive study, cited as ref. 32, compiles protein-protein interactions from various experimental studies, highlighting the complex regulatory mechanisms and diverse cellular functions in which ERRα is involved in different tissues.  

Regarding the second issue, also related to the first question of the Reviewer, “a major question is whether if their is organ specific differences in ERRα activity and energy metabolism”, the tissue-specific effects reported by the literature so far have been summarized in the various paragraphs. The take-home message emerging from this plentitude of studies is that, by improving mitochondrial activity, the LANCL/ERRα/PGC-1α axis allows each organ or tissue described in this Review to perform better in its specific function. As an example, increased mitochondrial respiration and oxidative metabolism allows skeletal muscle cells to endure physical activity and also provide more energy to tumor cells increasing their proliferative rate. It is not surprising that stimulation of mitochondrial respiration should have beneficial effects on whichever tissue-specific function occurs in that particular cell type. To improve clarity, as the Reviewer suggested, we also added information to Figure 2 regarding the described role of ERRα also in tumors and in the kidney.

Round 3

Reviewer 4 Report

Comments and Suggestions for Authors

I have no further concerns on the manuscript.